# Hyaluronic Acid-Based Nanoparticles for Protein Delivery: Systematic Examination of Microfluidic Production Conditions

**DOI:** 10.3390/pharmaceutics13101565

**Published:** 2021-09-26

**Authors:** Enrica Chiesa, Antonietta Greco, Federica Riva, Rossella Dorati, Bice Conti, Tiziana Modena, Ida Genta

**Affiliations:** 1Department of Surgery, Fondazione IRCCS Policlinico San Matteo, 27100 Pavia, Italy; 2Department of Drug Sciences, University of Pavia, 27100 Pavia, Italy; antonietta.greco@iusspavia.it (A.G.); rossella.dorati@unipv.it (R.D.); bice.conti@unipv.it (B.C.); tiziana.modena@unipv.it (T.M.); 3Department of Public Health, Experimental and Forensic Medicine, Histology and Embryology Unit, University of Pavia, 27100 Pavia, Italy; federica.riva01@unipv.it

**Keywords:** hyaluronic acid, hyaluronic acid-based nanocarriers, microfluidics, protein encapsulation efficiency, protein delivery, CD44 targeting

## Abstract

Hyaluronic acid-based nanoparticles (HA NPs) can be used to deliver a protein cargo to cells overexpressing HA receptors such as CD44 since they combine the low toxicity of the carrier and the retention of the protein integrity with the receptor-mediated internalization. HA properties play a crucial but sometimes unclear role in managing the formation and stability of the meshwork, cell interactions, and ultimately the protein entrapment efficacy. Nowadays, microfluidic is an innovative technology that allows to overcome limits linked to the NPs production, guaranteeing reproducibility and control of individual batches. Taking advantage of this technique, in this research work, the role of HA weight average molecular weight (Mw) in NPs formation inside a microfluidic device has been specifically faced. Based on the relationship between polymer Mw and solution viscosity, a methodological approach has been proposed to ensure critical quality attributes (size of 200 nm, PDI ≤ 0.3) to NPs made by HA with different Mw (280, 540, 710 and 820 kDa). The feasibility of the protein encapsulation was demonstrated by using Myoglobin, as a model neutral protein, with an encapsulation efficiency always higher than 50%. Lastly, all NPs samples were successfully internalized by CD44-expressing cells.

## 1. Introduction

Proteins-based therapies are crucial to mitigate and ameliorate incurable diseases [1,2,3]. Since proteins rule several essential functions in live beings, they are considered powerful therapeutic agents for a wide range of diseases with approximately 240 FDA approved proteins and peptides available [4,5,6]. In this regard, biotechnology is making great efforts to produce proteins that could be exploited for therapeutic and diagnostic purposes as well as for vaccines. However, if compared to small moieties, protein-based therapeutic agents are more sensitive to chemical-physical degradations during either the manufacturing or storage phase as well as in vivo delivery [7].

To overcome these issues, the formulation is a key point to be investigated to ensure high biocompatibility, payload protection, and proper biopharmaceutical behavior and/or targeting [8,9,10,11,12]. For this purpose, several nanosized drug delivery systems (nsDDSs) proved to be relevant for protein delivery [9,10,13]. They can decrease the renal clearance, improve accumulation into target tissue(s) as well as protect the payload from enzymatic degradation in circulation and extracellular space, and foster the intracellular uptake. Despite the highlighted promising outcomes, nsDDSs still showed some substantial limitations including low encapsulation efficiency (EE), which is strongly related to the nsDDSs composition and the manufacturing method employed, lack of a consistent product, and huge development costs [14,15].

Among several nsDDSs, polymeric nanoparticles (NPs) have stood out for their great ability to deliver therapeutic protein when systemically administered [3].

In the last years, hyaluronic acid (HA), discovered by K. Meyer and John W. Palmer in 1934 [16], has gained much attention due to its versatile applications in nanomedicine; modified or pure HA has been investigated to encapsulate proteins, antiseptics, antibiotics, and anticancer drugs, and to control their release [17]. HA is a hydrophilic, biodegradable, biocompatible, negatively-charged polysaccharide composed of D-glucuronic acid and N-acetyl-D-glucosamine repeated units linked via glycosidic bonds (β-1→4 and β1→3) [18,19,20]. HA is widespread in the human body, and it is involved in many structural and biological activities. Its biological role is dramatically influenced by the weight average molecular weight (Mw). HA can be a heat shock proteins inducer (0.4–4.0 kDa Mw), it can stimulate the immune response or angiogenesis (6–20 kDa Mw), it takes part in wound healing and ovulation (20–200 kDa Mw), and it can limit the angiogenesis and the immunologic response (Mw > 500 kDa) [21]. Moreover, HA is the natural ligand of the cluster of differentiation-44 (CD44) receptors, transmembrane glycoproteins overexpressed in pathological tissues, and correlates with the cancer progression, metastasis, and anticancer drug resistance [22,23]. With regards to HA-based NPs (HA NPs), it is also known that HA has a role as a non-adhesive polymer toward most blood proteins and is useful in prolonging NPs circulation time when systemically administered by preventing opsonization [24,25].

HA NPs were traditionally obtained by bulk methods based on ionotropic gelation complexation mechanisms due to the polyelectrolytic interaction between oppositely charged polymers among which chitosan (CS), a polysaccharide constituted by glucosamine and N-acetylglucosamine, represents one of the most promising candidates for biomedical applications [26,27,28,29,30,31,32,33]. Ionotropic gelation technique between HA and CS is mainly attractive for proteins/peptides encapsulation because it is organic solvent-free, it uses compatible pH or ionic strength, and it does not expose proteins to high temperature or pressure, and so it permits to successfully encapsulate macromolecular payloads into meshwork, ensuring a controlled release while retaining their structural integrity [34,35,36]. However, the lack of control over the mixing causes low batch-to-batch reproducibility, slow production rate, and low encapsulation efficiency [37,38,39]. In the use of NPs made by electrostatic complexation, the moieties properties have a key but sometimes uncertain role in controlling the physical properties and stability of the complex, their biological performance, and ultimately the payload loading. Different from other types of nsDDSs where ordered structures can be found and characterized, the components of electrostatic complexes are likely to be disordered, and therefore it may be assumed that the carrier performance can be tuned solely by controlling the interaction between HA and CS. The two polymers’ chains may be influenced by steric hindrance and proximity of charged groups regulated through the degree of acid/base dissociation and the polymer molecular weight. Several other variables further confuse the scenario, with differences in the preparative methods and, unsurprisingly, literature is rife with conflicting evidence on HA NPs made by HA with different Mw and various preparative procedures [40,41,42,43,44].

The microfluidic technique has been demonstrated as a promising and versatile technology capable of overcoming nsDDS manufacturing drawbacks [45,46] to obtain lipid and polymer-based NPs with high batch-to-batch reproducibility, precise modulation of the NPs size, improved drug encapsulation efficiency (EE), and easy scale-up [47,48,49,50]. Microfluidic technology guarantees fast and easy production of high-quality NPs by virtue of the controlled manipulation of small volumes of reagents into micrometric channels [46,47]. In a previous study, we set up a novel one-step microfluidics-based method for the preparation of HA NPs in which ionotropic gelation between CS and sodium tripolyphosphate (TPP) and HA/CS polyelectrolytic complexation were simultaneously exploited to obtain tailored HA NPs [51]. Based on our expertise in the field, the aim of this work is to propose a methodological approach to produce highly protein-loaded HA NPs with comparable physical characteristics starting from HAs with different Mw (280 kDa, 540 kDa, 710 kDa and 820 kDa). To the best of our knowledge, this is the first attempt to manage HA-based NPs production through a microfluidic device starting from different HAs but exploiting the same process parameters without impacting the formulation characteristics. This approach could be useful for rapid product development and an easier scale-up.

The feasibility and reliability of the one-step microfluidics-based preparation method was firstly investigated by evaluating the effect of the HA Mw on NPs physical characteristics (size, polydispersity index (PDI) and surface charge) and then formulation parameters were duly amended, maintaining constant process parameters to obtain NPs as similar as possible regardless of the HA Mw.

Myoglobin (Myo), a globular protein of MW 17,600 g/mol, was used as a neutral reference protein (pI: 6.8–7.4) to evaluate the potential loading of a macromolecular drug not supported by polymer–drug ionic interaction.

Myo-loaded NPs (Myo-HA NPs) were synthetized using the set-up formulation parameters for each HA tested and evaluating different Myo:CS weight ratios in the range 1:5–1:20. Myo encapsulation efficiency was determined by UV-vis spectrometry [52]. Selected Myo-HA NPs batches were then analyzed in relation to the macromolecular drug release.

Furthermore, cytocompatibility of HA NPs was assessed on human mesenchymal stem cells (hMSCs) from bone marrow. Preliminary uptake studies were also carried out on the same cell line where CD44 expression was confirmed.

## 2. Materials and Methods

### 2.1. Materials

Hyaluronic acid sodium salt (HA, weight average molecular weight (Mw) 710 kDa), sodium tripolyphosphate (TPP) and Myoglobin (Myo) from equine heart were purchased from Sigma Aldrich (St. Louis, MO, USA). HAs (Mw 280, 540, and 820 kDa) were from Faravelli SpA (Milan, Italy). Chitosan chloride salt pharmaceutical grade (CS, Chitoceuticals, viscosity 19 mPa (1% in water), deacetylation degree 82.2%, chloride content 13%) was obtained from Heppe Medical ChitosanGmbH (Halle, Germany). Thiazolyl Blue Tetrazolium Bromide (MTT, approx. 98% TLC), Dulbecco’s Modified Eagle’s Medium High glucose (DMEM), Dulbecco’s Phosphate Buffered Saline (PBS 10×X, sterile), Trypsin-EDTA, Dimethyl sulfoxide (DMSO), and Penicillin-Streptomycin were supplied from Sigma Aldrich (St. Louis, MO, USA). Fetal Bovine Serum was from EuroClone Spa (Milan, Italy). Unless specified the water used was distilled and filtered through 0.22 μm membrane filters (Millipore Corporation, Billerica, MA, USA) and all the solvents used were of HPLC or analytical grade.

### 2.2. Cell Lines

hMSCs from human bone marrow were kindly offered by the Department of Public Health, Experimental Medicine and Forensic, Histology and Embryology Unit, University of Pavia.

### 2.3. HA NPs Preparation Method

NPs were synthetized by using the automated mixing microfluidic platform NanoAssemblr^TM^ Benchtop (Precision NanoSystems Inc., Vancouver, BC, Canada) equipped with a staggered herringbone micromixer (SHM) previously described in [51,53]. The Y-shaped inlet channel allows the pumping of two fluids into the SHM device separately: HA and TPP dissolved in water at appropriate concentrations and CS aqueous solution (Table 1).

Based on the experimental parameters previously set-up for 710 kDa Mw HA [51], the HA, CS, and TPP concentrations were set at 0.150 mg/mL, 0.050 mg/mL, and 2 µg/mL, respectively.

Using HA of different Mw, HA solutions concentration was suitably modified as a function on the HA Mw used by multiplying the starting concentration used for 710 kD Mw HA (0.150 mg/mL) by the ratio between 710 kD and the Mw of the newly used HA (280, 540, or 820 kD, respectively); CS concentration was duly varied, maintaining the previously selected HA:CS weight ratio (3:1 *w*:*w*). Polymeric solution concentrations used are summarized in Table 1.

The concentration of TPP, used as CS’s ancillary cross-linker, was kept constant at 2 μg/mL.

Microfluidics process parameters were used as set up in [51]: a 1:1 flow rate ratio (FRR) and a total flow rate (TFR) of 12 mL/min. A sample of 3 mL was recovered from the exit point without the initial and the ended waste (respectively set at 0.350 mL and 0.050 mL).

### 2.4. Rheological Characterization of HA Solutions

The rheological behaviour of HA and CS solutions were analysed by the Rheometer Kinexus Plus (Malvern, Alfatest, Milan, Italy) with a circulated environmental system for temperature control. The sample was loaded between cone-plate geometry CP1/60:PL61 with a gap of 1 mm. Data processing was recorded with rSpace software. For appropriate prompt information regarding the rheological behaviour, the table of shear rates/equilibrium flow curve analysis was carried out at 25 °C: This sequence runs a logarithmic table of shear rates and measures the viscosity by increasing the shear rate values in the range 0.1–100 s^−1^, 10 samples per decade. The viscosity value was plotted as mean ± standard deviation (SD) of three independent measurements.

### 2.5. Myo Loading in HA NPs

Myo, the chosen model neutral protein, was added to CS aqueous solution with different Myo-to-CS weight ratios, namely 1:5, 1:10, 1:15, and 1:20 (*w*:*w*).

Both Myo-CS and HA-TPP aqueous solutions were separately injected through the two SHM microchannels according to the process conditions described in Section 2.3. All produced Myo-loaded HA (Myo-HA) samples are summarized in Table 2.

### 2.6. HA NPs Characterization

The mean diameter, PDI, and surface charge of both placebo NPs and Myo-loaded HA NPs were measured at room temperature by dynamic light scattering (DLS) by using NICOMP 380 ZLS (Particles Sizing System, Santa Barbara, CA, USA). All the analyses were carried out in triplicate for each formulation and results reported as mean ± SD. Morphology was evaluated by transmission electron microscopy (TEM) (JEOL JEM-1200EXIII with TEM CCD camera Mega View III, Tokyo, Japan) with negative staining of 1% (*w*/*v*) uranyl acetate; the images obtained were processed by ImageJ software (NIH, Bethesda, MD, USA) [54] to further investigate the NPs sizes.

Myo encapsulation into NPs was detected by UV-vis spectrophotometer (6705 Model, Jenway, Staffordshire, UK) at the wavelength of 409 nm. First, Myo-HA NPs were centrifugated (16,400 rpm, 4 °C, 30 min) (Eppendorf Centrifuge 5417 R, Eppendorf s.r.l., Milan, Italy) and the recovered supernatants were analyzed. Myo content was determined against the calibration curve derived from Myo standard solutions in a concentration range 1–100 µg/mL (R^2^ = 0.999). The outcomes were expressed as percentage value (EE %) ± SD calculated from three independent experiments by using the following Equation (1):EE % = (Myo_w_ − Myo_sup_)/(Myo_w_) × 100(1)
where the Myo_w_ is the drug amount used for the NPs batch synthesis, and Myo_sup_ is the drug detected in the supernatant.

The drug release evaluation was performed on selected Myo-HA NPs batches. Myo-HA NPs batches were centrifugated (16,400 rpm, 4 °C, 30 min) and collected NPs were resuspended in 0.01 M PBS pH 7.4 at the concentration of 60 µg/mL and incubated at 37 °C. At scheduled time points, samples were centrifuged (16,400 rpm, 4 °C, 30 min) and supernatants were analyzed by UV–vis spectrophotometer according to the method above reported. Results were expressed as mean drug release percentage ± SD (experiments carried out in triplicate). The Myo release data were plotted as (i) a cumulative amount of the drug released versus the time (zero order kinetics model); (ii) cumulative percentage of the drug released versus the time (first-order kinetics model); and (iii) a cumulative percentage of the drug release versus the square root of time (Higuchi kinetic model). Moreover, the Myo release data up to 60% were analyzed by the empirical Ritger and Peppas equation.

Moreover, potential swelling phenomena were investigated checking NPs mean size by DLS at each scheduled release time point.

### 2.7. Cytotoxicity Assay

To investigate the toxicity of placebo HA NPs, MTT (3-[4,5-dimethylthiazol-2-yl]-2,5-diphenyltetrazolium bromide) assay was performed on hMSCs, cultured in DMEM with 10% FBS and 1% Penicillin-Streptomycin solution. hMSCs were chosen as model cell lines due to the well-established CD44 expression [55]. hMSCs were seeded in 96-well plates and incubated overnight to reach the density of 10,000 cells. Subsequently, cells were treated with different concentrations of HA NPs (12.5–450 µg/mL) for 24 h. Following the incubation time, the MTT test was carried out as described in [51]. The viability was detected through the absorbance values measured at 570/690 nm (SpectraMax M2e, Molecular Device LLC, San Jose, CA, USA).

### 2.8. Uptake Studies

For uptake studies, fluorescent HA NPs were prepared. Rhodamine B (RhB), a well-known fluorescent dye, was chemically grafted to CS (CS-RhB) through an amidation reaction as in [51]. Specifically, fluorescent HA (HA-RhB) NPs were produced as set up above by using CS solution composed of plain CS (90% *w*/*w*) and CS-RhB (10% *w*/*w*).

All HA-RhB NPs samples were characterized in size, PDI, and charge surface by DLS. Results were shown as mean ± SD (measurements carried out in triplicate).

Cellular uptake of HA-RhB NPs was investigated by confocal microscopy (Leica TCS SP8 with AOBS^®^ beam splitting device, Leica Microsystems, Wetzlar, Germany). hMSCs were seeded on the bottom glass slide (20,000 cells/well) and cultured (at 37 °C, 5% CO_2_) in DMEM with FBS (10% *v*/*v*) and antibiotics (1% *v*/*v*) until reaching 80% of cells confluence. Subsequently, HA-RhB NPs (100 µg/mL) were added and incubated for 90 min as selected from our previous work [51]. At the end of the incubation, all culture media were discarded and hMSCs were washed at least three times with sterile PBS buffer. Finally, hMSCs were fixed with 4% (w) paraformaldehyde. CD44 expression was figured out through immunocytochemistry assay, as reported in [51]. Cell nuclei were stained by Hoechst33258 (Sigma Aldrich, St. Louis, MO, USA). Cells were observed by confocal microscopy (obj mag 40×).

### 2.9. Statistical Analysis

Results are represented as mean ± SD of at least three independent batches. The statistical significances were assessed by ANOVA tests, with Tukey’s multiple comparison test. All the statistical evaluation were performed in GraphPad Prism version 6 (GraphPad Software Inc., La Jolla, CA, USA).

## 3. Results and Discussion

### 3.1. NPs Preparation and Characterization

Preliminary experiments were carried out to evaluate the effect of HA Mw on NPs dimensional parameters (mean size and size distribution) by using the experimental conditions previously set up by our research group [51] when a microfluidics-assisted one-step manufacturing method was developed for 710 kDa Mw HA-based NPs, here used as the reference sample. Briefly, HA solution (0.15 mg/mL) and CS solution (0.05 mg/mL) were pumped at 12 mL/min into the SHM microfluidic device with a FRR of 1:1. HAs with different Mw were tested, namely 280 kDa, 540 kDa, 710 kDa, and 820 KDa.

As expected, the reference sample (HA_710) showed a mean size of 200 nm with a satisfactory PDI of 0.3 and negative surface charge of about −20 mV (Table 3). Instead, heterogeneous NPs (PDI > 0.5) with mean diameter higher than 600 nm were obtained by using HA of 280 kDa and 820 kDa. For HA with Mw of 540 kDa, NPs with mean size of 306.5 ± 36.9 nm (PDI = 0.416 ± 0.074) were recovered. The disagreement between results could be ascribed to the change in viscosity of the HA solution depending on the HA Mw. As already demonstrated for other microfluidic devices [56], the ratio of the dispersed phase viscosity to that of the continuous phase is a critical parameter for microfluidic two-phase flows and a variation in the viscosity of one of the mixing liquids leads to a shift of the flow patterns inside the microchannel while generally the flow regimes are conserved. In order to verify this behaviour in the SHM microchannel, the viscosity of the HA solutions employed was experimentally assessed. Figure 1 shows the equilibrium flow curves of HA solutions obtained using HA at different Mw and keeping constant the concentration at 0.15 mg/mL. As forecast, at low shear rate the viscosity notably depends on the HA Mw. The highest Mw tested (820 kDa) revealed the highest viscosity ranging from 0.26 to 0.02 Pa × s for a shear rate of 0.1–1 s^−1^. In the same shear rate range, a clear decrease of viscosity was observed for 280 kDa Mw HA solution, always lower than 0.01 Pa × s. Similar viscosity profiles, in a rank of 0.05 Pa × s–0.01 Pa × s for share rate of 0.1–1 s^−1^, were highlighted for HA solutions prepared with 540 kDa and 710 kDa Mw HAs.

Knowing how to manufacture HA-based NPs by using different raw polymers but the same process parameters without impacting formulation characteristics could be advantageous for a rapid product development and an easier scale-up [57,58,59].

In the attempt to demonstrate our microfluidics-based method versatility and reliability in HA NPs manufacturing, formulation parameters were duly amended to obtain HA NPs made of different HAs while proper physico-chemical characteristics were maintained: size of around 200 nm, ideal for the optimum drug delivery [60]; PDI values always lower than 0.3, indicating a good sample homogeneity; and negative ζ potential proving the peripheric HA distribution on the NPs, which is an essential factor for the active targeting to CD44 receptors [36,61]. As a function of the HA Mw, the HA solutions concentration was modified (as described in Section 2.3, Table 1) while HA:CS weight ratio was kept at 3:1 *w*:*w* in order to provide the same proper HA/CS electrostatic interaction; CS concentrations were suitably changed, as shown in Table 1. A constant negligible amount of TPP was employed with ancillary function, as a cross-linker to physically trap CS, resulting in controlled gelation of CS in the form of spherical and homogeneous NPs [35,43,62]. An understanding of the mechanisms of flow pattern formation inside the microchannel could be crucial for the design of tailor-made nanoformulations; dynamic viscosities of the different HA solutions used for the preparation of HA_280/_540/_710/_820 NPs were measured. The equilibrium flow curves of the different solutions used are reported in Figure 2. The variation of HA concentrations at 0.325, 0.200, and 0.130 mg/mL, by using HA with Mw of 280 kDa, 540 kDa and 820 kDa, respectively, lead to superimposable viscosity profiles with respect to that of the 0.150 mg/mL 710 Mw HA solution (reference sample).

Moreover, changing the CS solution concentration from 0.125 to 0.043 (Table 1) did not affect the solution viscosity that ranged from 0.03 to 0.01 Pa × s for the share rate of 0.1–1 s^−1^ for all the CS solutions tested.

NPs batches were then characterized and the outcomes overview is reported in Table 3. HA_710 NPs (reference sample) confirmed a mean particles size of 200.22 ± 28.97 nm, PDI of 0.30, and ζ potential of −16.72 ± 2.28 mV. HA_280 NPs revealed a mean size of 211.32 ± 19.25 nm, PDI of 0.3, and a negative surface charge of −16.58 ± 2.21 mV. Increasing the HA Mw at 540 kDa, HA_540 NPs size was 136.90 ± 20.19 nm with a PDI of 0.3 whilst the measured NPs charge was still negative (−19.70 ± 5.35 mV). Finally for the highest Mw tested (820 kDa), DLS analysis revealed NPs of 197.58 ± 27.14 nm, acceptable size uniformity (PDI = 0.29), and negative surface charge (−19.53 ± 2.16 mV).

Statistical analysis performed by the Tukey’s multiple comparisons test revealed no significant differences among NPs samples produced starting from different Mw HAs (*p* < 0.05).

Therefore, experimental data would account for the impact of the fluid viscosity on the mixing performance into the microfluidic SHM device and on polymers aggregation in nanosized forms, suggesting that the control of fluid viscosity may tune the HA/CS assembly into the microchannel. In particular, HA solutions characterized by comparable viscosities, even if HAs with different Mw are used, allow for the formation of NPs with similar physical properties using the same process parameters.

Knowing the relationship between the logarithm of the polymer Mw and its solution viscosity [63,64,65,66], as well as the superimposable viscosity values assessed for the different HA solutions employed (Figure 2), HA Mw and the chosen experimental HA solutions concentrations were plotted and a logarithmic interpolation was found (R^2^ of 0.9886) (Figure 3).

The calculated linear relationship (R^2^ of 0.9886) permits to estimate, with a good approximation, the HA solution concentration to be used for a HA of specific Mw to obtain NPs with the selected size (~200 nm), PDI (<0.3), and negative surface charge using unchanged process parameters. It is worth pointing out that we have for the first time identified a methodological approach to produce HA-based NPs starting from different HAs but exploiting the same process parameters without impacting the formulation characteristics. This approach can positively impact either the time or cost of a new formulation development and its scale-up feasibility.

At the end, the overall shape and surface morphology of NPs were analyzed through TEM. Results, reported in Figure 4, showed a general spherical shape for all Mw HA NPs. TEM images were elaborated by ImageJ software finding HA NPs mean size of around 200 nm in agreement with the diameter results detected by DLS measurements.

### 3.2. Myo Loading in HA NPs

As a rule of thumb, ionic interactions between polymers and macromolecular drugs are largely exploited to increase NPs drug loading. In this study, Myo was chosen for mimicking the most challenging drug entrapment case (*worst case*): NPs synthesis takes place in water at neutral pH in which the protein has no charged groups (6.8–7.4 pI), and no strong ionic interactions can occur between HA or CS and Myo.

Myo-loaded HA NPs batches were prepared testing four different Myo-to-CS weight ratios, namely 1:5, 1:10, 1:15, and 1:20 (*w*:*w*). Produced NPs were characterized for particle size, size distribution, and ζ Potential, in the same conditions of placebo formulations. Results (Table 4) demonstrated a slight but not significant increase of NPs dimensions with respect to relative placebo NPs (*p* value > 0.05, by Tukey’s multiple comparisons test) and a slight decrease of the size uniformity (PDI ≥ 0.3), attributable to the Myo incorporation into the polymers mash. Moreover, the highest Myo-to-CS weight ratio used (1:5), corresponding to the highest Myo amount loaded into NPs, caused the most relevant NPs enlargement in respect to the empty formulations, and an important increase of NPs heterogenicity with PDI values always higher than 0.35, for all HA Mws tested. Nevertheless, statistical analysis demonstrated negligible differences among Myo-loaded NPs samples and the relative placebo formulations.

Regardless of payload, all formulations showed negative ζ Potential values, suggesting that HA was still distributed on NPs external surface and thus the protein entrapment into the polymeric matrix may be assumed.

Regarding the ability of NPs to effectively encapsulate proteins, the low payload is always the limiting condition in product development from both a therapeutic and economic point of view [9,50], hence suitable manufacturing technologies should be developed to achieve a consistent product.

Myo amount encapsulated into HA NPs was assessed by spectrophotometric analysis at 409 nm wavelength. Myo amounts and the relative EE% obtained were listed in Table 4. All the formulations showed a rather satisfactory EE% higher than 53%. Neither Mw HA nor Myo:CS ratio impacted on the amount of encapsulated macromolecular drug, revealing a similar and very satisfactory average EE% (calculated from the EE% of the different Myo:CS for each HA Mw tested) of 77.28, 74.33, 73.15, and 71.68%, respectively, for 280 kDa, 540 kDa, 710 kDa, and 820 kDa HA. Considering Myo’s pI and its lack of charge in the neutral environment in which NPs have been synthesized, ionic interaction among Myo/HA/CS can be excluded, and the high EE% outcomes could be attributed to the microfluidic technique, revealing a suitable method to produce consistent samples.

To investigate the Myo release, the best batch in terms of size distribution (PDI < 0.3) and protein EE% (EE% > 65%) was selected for each HA Mw used: (I) Myo280_20; (II) Myo540_15; (III) Myo710_15; and (IV) Myo820_20 (Table 4—highlighted in grey). Briefly, Myo in vitro release tests were performed in 0.01 M PBS pH 7.4 at a constant temperature (37 °C). Figure 5A reports the Myo release profiles from the selected batches. The drug release was monitored for 72 h. Myo540_15 released all the loaded model protein in 12 h of incubation; all the other formulations prolonged the Myo release till 24 h of incubation. The significantly fastest Myo release from Myo540_15 is attributable to the smaller NPs sizes (mean diameter around 130 nm) [67]. NPs made of HA with the highest Mws (820 and 710 kD) and similar dimensions (mean size around 210–250 nm) showed comparable Myo release profiles. The slowest Myo release profile was detected from NPs made of HA with the lowest Mw (mean diameter around 250 nm), This behaviour can be ascribed to the higher flexibility of HA chains able to more tightly interact with CS to form NPs with compact structure [68]. The order of drug release kinetics was determined by plotting the cumulative concentration of drug against time and square root of time. The model that best fits the Myo release data was evaluated by correlation coefficients (R^2^) [69,70]. The higher correlation coefficient was revealed for the Higuchi model for all the formulations tested (R^2^ > 0.95), demonstrating that the Myo release from the HA/CS NPs’ matrix is diffusion-controlled. Finally, by fitting the Myo release data to the Korsmeyer–Peppas model equation [71], the diffusion exponent ranged from 0.51 to 0.70, indicating, for a sphere geometry, an anomalous diffusion mechanism.

In order to evaluate HA NPs physical stability in PBS at neutral pH, NPs mean sizes were further measured at each scheduled release time point and swelling phenomena are illustrated in Figure 5B. A NPs enlargement of around 50–70 nm was observed for all polymers used by 6 h; additionally, NPs sizes remained below 300 nm after 24 h, suggesting that NPs are rather stable in simulated physiological conditions.

### 3.3. In Vitro Cytotoxicity of HA NPs

HA and CS have been reported to be biodegradable and biocompatible polymers, making them ideal candidates to make safe DDS [17,27]. Since the morphology, size, and surface charge of the NPs may affect their toxic impact on the cells, cytocompatibility of all the placebo nano-formulations were investigated by MTT assay at the NPs concentration range 12.5–450 µg/mL. Results (Figure 6) confirmed the well-known biocompatibility of the formulations, highlighting that the cell viability percentages are always higher than 80% regardless of the HA Mw; moreover, in this work, the low HA NPs toxicity was also guaranteed by the preparation method that exclusively used water as a solvent as well as a suitable pH for the final formulation (around 7.4). Unlike HA/CS_540, other HA-based NPs slightly affected the biological and biochemical cellular functions, causing a cell viability reduction of around 20% along with increasing the NPs concentration from 12.5 to 450 µg/mL.

All the outcomes obtained were supported by the morphological evaluations of the cells treated, which maintained a healthy conformation and a great distribution on the plate comparable to the untreated cells (CTRL).

### 3.4. In Vitro HA NPs Uptake by hMSCs

To perform the best therapeutic activities, nsDDSs need to deliver and interact with specific targets cells [60]. As is widely known in the literature, the nanocarrier physicochemical properties could dramatically influence target cells uptake [67,72,73].

In this study, we preliminary investigated whether all the HA NPs could smoothly interact with the model cells hMSCs, which overexpress CD44 receptors, analyzing simultaneously the NPs’ active targeting. To follow the NPs’ behaviors in the cells, fluorescent HA NPs (HA-RhB NPs), using HA with different Mw, were synthetized. All HA-RhB NPs samples obtained were characterized in terms of size, PDI, and charge surface, showing negligible differences when compared with not-fluorescent HA NPs (Table 5).

After the incubation of 20,000 hMSCs, with 20 µg of HA-RhB NPs for 90 min, the confocal microscope images (Figure 7) displayed that all the HA-RhB NPs massively crossed the hMSCs plasmatic membranes and are chiefly located in the cytoplasmatic compartment. All the cells presented normal morphology, by further confirming the cytocompatibility of NPs. Hence, it is possible to confirm that HA-based NPs regardless of the HA Mw can comfortably interact, in 90 min, with hMSCs without affecting the cells’ behaviors.

## 4. Conclusions

Taking advantage of a previously set-up microfluidics technique, in this work we proposed a methodological approach to produce HA-based NPs by using different raw polymers but with the same process parameters without impacting formulation characteristics. A mathematical relationship between HA Mw and HA solution concentration used in NPs manufacturing was established, and it can be exploited for easier formulation prototyping and faster scale-up.

HA-based NPs made by ionotropic gelation are promising for protein delivery since they were produced in a solvent-free environment, at compatible pH or ionic strength, and without exposing the protein to high temperature or pressure. However, the polymer features (Mw, number and proximity of charged group, and steric hindrance) play an important but not well-understood role in controlling NPs formation and stability, payload encapsulation, and intracellular uptake.

This microfluidics-based method ensures NPs made of different Mw HAs can be produced with the same critical quality attributes using the same process parameters, thus providing a direct and rapid production pathway for nanomedicine from a bench to a prototype product. The possibility to effectively load proteins (as confirmed by Myo encapsulation and release studies, *worst case*) highlights the possible application of the developed formulations for protein-based therapies.

Finally, based on the preliminary investigation of the NPs cellular internalization, further studies will be undertaken to elicit a deep understanding of whether the different HA Mw could select a specific endocytosis pathway to be internalized in the cells.

## Figures and Tables

**Figure 1 pharmaceutics-13-01565-f001:**
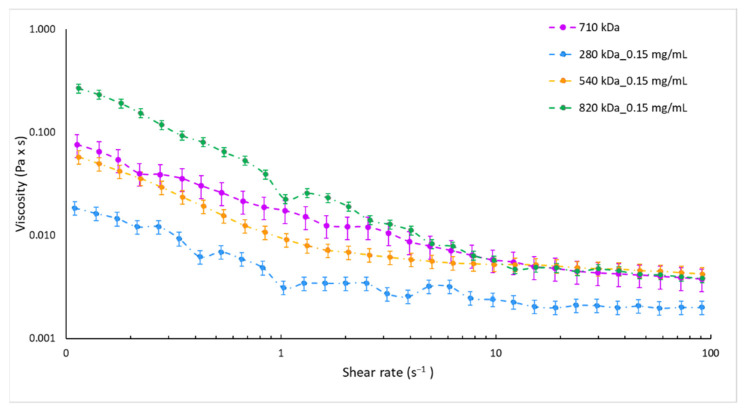
Equilibrium flow curves of 0.15 mg/mL HA aqueous solutions by HA with different molecular weights, namely 280 kDa (blue dotted line), 540 kDa (orange dotted line), 710 kDa (violet dotted line), and 820 kDa (green dotted line).

**Figure 2 pharmaceutics-13-01565-f002:**
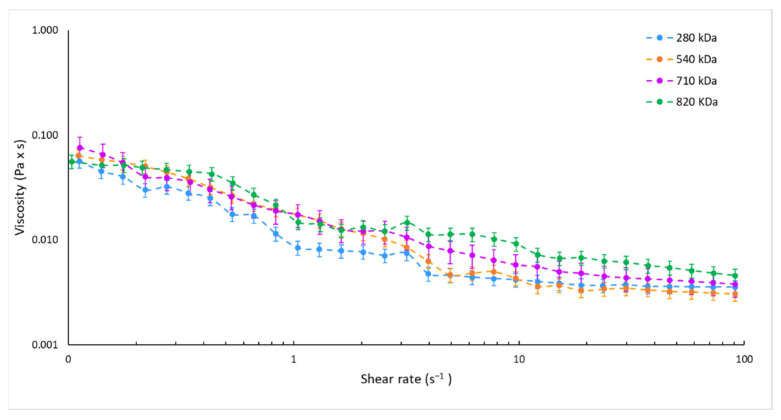
Equilibrium flow curves of HA solutions at 0.325, 0.200, 0.150, and 0.130 mg/mL by using HA with Mw of 280 kDa (blue dotted line), 540 kDa (orange dotted line), 710 kDa (violet dotted line), and 820 kDa (green dotted line), respectively.

**Figure 3 pharmaceutics-13-01565-f003:**
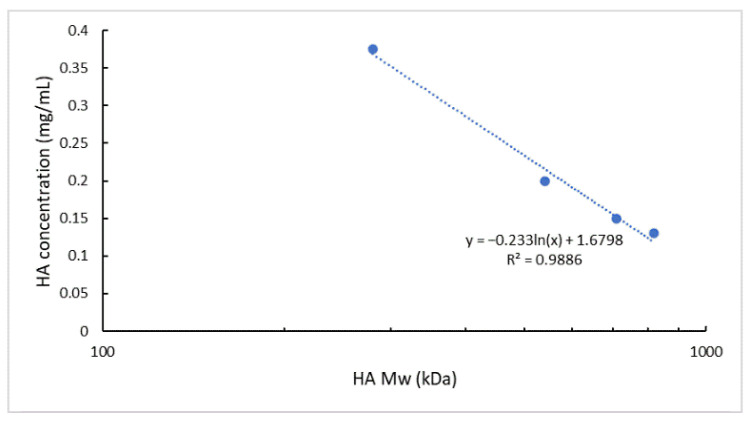
Interpolation of the ln (HA Mw) vs. HA solution concentration. Experimental points correspond to 280, 540, 710, and 820 kDa Mw HA and the relative HA solution concentrations used in the NPs microfluidics-based preparation method.

**Figure 4 pharmaceutics-13-01565-f004:**
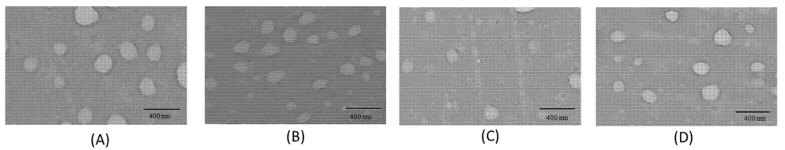
TEM images of HA NPs: (**A**) HA_280 NPs; (**B**) HA_540 NPs; (**C**) HA_710 NPs; (**D**) HA_820 NPs. (Scale bar = 400 nm).

**Figure 5 pharmaceutics-13-01565-f005:**
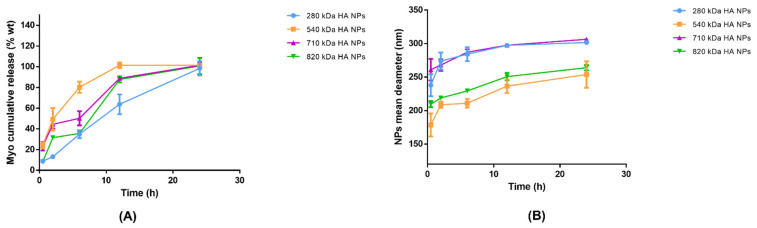
(**A**) Myo in vitro release from Myo280_20 (blue line), Myo540_15 (orange line), Myo710_15 (violet line), and Myo820_20 (green line) in 72 h: Myo-HA NPs were incubated in 0.01 M PBS pH 7.4 at 37 °C to best simulate the physiologic environment. Results represent the mean ± SD; *n* = 3 independent batches. (**B**) Swelling phenomena representation: Myo280_20 (blue line), Myo540_15 (orange line), Myo710_15 (violet line), and Myo820_20 (green line).

**Figure 6 pharmaceutics-13-01565-f006:**
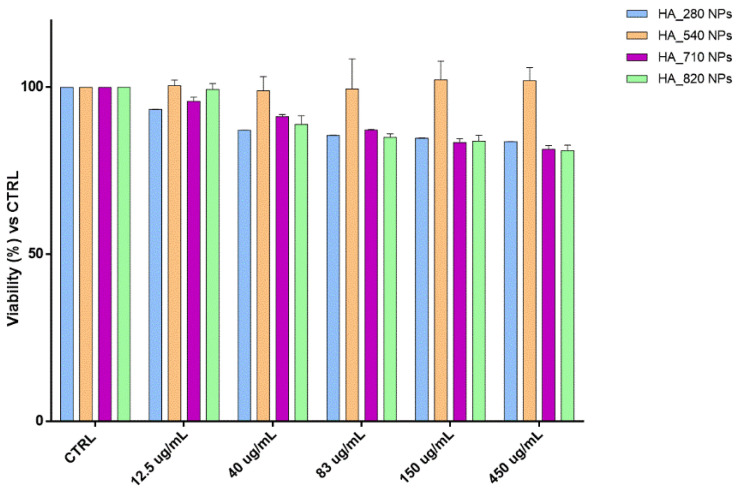
Cytotoxicity outcomes from the incubation of hMSCs and increasing HA NPs concentration. The graph shows the cell viability percentage compared with the viability of untreated cells (CTRL).

**Figure 7 pharmaceutics-13-01565-f007:**
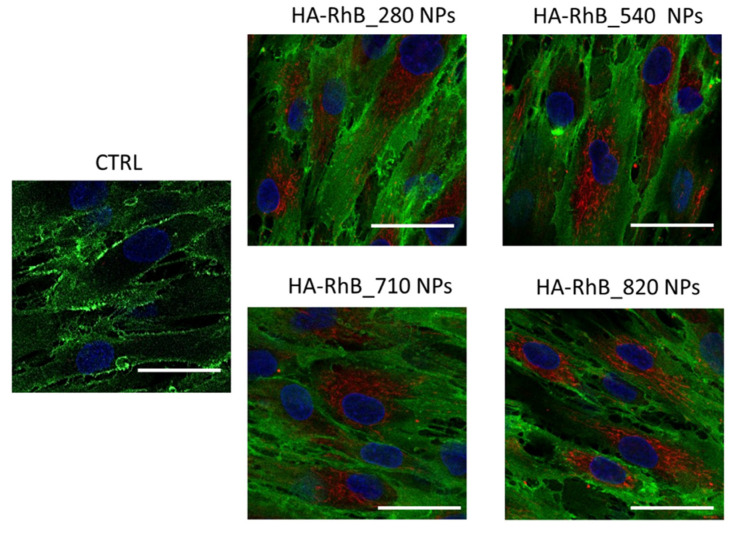
Confocal microscopy images of hMSCs incubated with 20 µg of HA-RhB NPs (for each HA Mw) for 90 min; CTRL represent untreated hMSCs. Red fluorescence represents HA NPs; nuclear blue fluorescence represent DNA with Hoechst33258 dye; hMSCs membrane was labelled by using anti-CD44 primary antibody and FITC-labelled secondary antibody (green fluorescence); scale bar = 20 µm.

**Table 1 pharmaceutics-13-01565-t001:** HA and CS solution concentrations used to synthetize NPs.

* Placebo NPs Sample Code	HA Mw (kDa)	[HA] mg/mL	[CS] mg/mL
HA_280	280	0.375	0.125
HA_540	540	0.200	0.067
HA_710	710	0.150	0.05
HA_820	820	0.130	0.043

* All samples were prepared almost in triplicate.

**Table 2 pharmaceutics-13-01565-t002:** Summary table of Myo-loaded HA NPs.

HA Mw (kDa)	* Sample Code	Myo:CS (*w*:*w*)
280	Myo280_5	1:5
	Myo280_10	1:10
	Myo280_15	1:15
	Myo280_20	1:20
540	Myo540_5	1:5
	Myo540_10	1:10
	Myo540_15	1:15
	Myo540_20	1:20
710	Myo710_5	1:5
	Myo710_10	1:10
	Myo710_15	1:15
	Myo710_20	1:20
820	Myo820_5	1:5
	Myo820_10	1:10
	Myo820_15	1:15
	Myo820_20	1:20

* All samples were prepared almost in triplicate.

**Table 3 pharmaceutics-13-01565-t003:** Characterization of different Mw HA NPs. Outcomes represent the mean ± SD; *n* = 3 independent batches.

Sample	Diameter ± SD (nm)	PDI ± SD	ζ Potential ± SD (mV)
HA_280	211.32 ± 19.25	0.30 ± 0.02	−16.58 ± 2.21
HA_540	136.90 ± 20.19	0.27 ± 0.01	−19.70 ± 5.35
HA_710	200.22 ± 28.97	0.30 ± 0.03	−16.72 ± 2.28
HA_820	197.58 ± 27.14	0.29 ± 0.01	−19.53 ± 2.16

**Table 4 pharmaceutics-13-01565-t004:** Characterization of Myo-HA NPs. Results represent the mean ± SD; *n* = 3 independent batches. Samples selected for further studies were highlighted in grey.

Sample Code	Diameter ± SD (nm)	PDI ± SD	ζ Potential ± SD (mV)	Myo Amount ± SD (µg)	EE% ± SD
Myo280_5	326.37 ± 95.68	0.43 ± 0.01	−17.99 ± 3.93	26.01 ± 0.46	80.06 ± 1.43
Myo280_10	303.07 ± 37.15	0.39 ± 0.01	−17.74 ± 5.37	13.48 ± 0.06	82.97 ± 0.4
Myo280_15	308.73 ± 81.53	0.42 ± 0.05	−18.00 ± 4.59	8.14 ± 0.23	75.15 ± 2.14
Myo280_20	245.30 ± 3.45 ^ns^	0.28 ± 0.03	−19.34 ± 8.37	5.75 ± 0.32	70.97 ± 3.39
Myo540_5	164.43 ± 7.01	0.35 ± 0.09	−17.63 ± 2.61	8.48 ± 0.11	85.90 ± 1.13
Myo540_10	158.75 ± 18.89	0.27 ± 0.12	−18.93 ± 9.69	4.25 ± 0.06	53.05 ± 0.65
Myo540_15	157.36 ± 16.84 ^ns^	0.30 ± 0.04	−14.38 ± 7.70	2.82 ± 0.13	85.90 ± 3.91
Myo540_20	140.8 ± 29.63	0.37 ± 0.02	−16.17 ± 6.59	1.93 ± 0.06	78.20 ± 2.60
Myo710_5	463.00 ± 15.87	0.53 ± 0.05	−20.00 ± 4.72	10.42 ± 0.06	80.15 ± 0.49
Myo710_10	412.33 ± 67.68	0.53 ± 0.05	−12.26 ± 3.76	5.63 ± 0.25	86.57 ± 3.93
Myo710_15	268.40 ± 23.13 ^ns^	0.30 ± 0.01	−19.65 ± 2.93	3.01 ± 0.17	69.57 ± 3.92
Myo710_20	261.33 ± 56.72	0.55 ± 0.07	−19.47 ± 4.38	0.85 ± 0.40	56.29 ± 12.36
Myo820_5	520.32 ± 120.66	0.47 ± 0.04	−10.90 ± 2.91	7.11 ± 0.11	63.63 ± 1.00
Myo820_10	468.18 ± 101.66	0.46 ± 0.05	−19.52 ± 3.82	3.38 ± 0.28	60.47 ± 5.02
Myo820_15	211.50 ± 23.06	0.43 ± 0.07	−14.96 ± 5.21	3.44 ± 0.23	92.53 ± 0.01
Myo820_20	211.87 ± 34.96 ^ns^	0.31 ± 0.09	−14.3 ± 4.55	1.96 ± 0.01	70.10 ± 0.02

^ns^ indicates the non-significant increase of selected Myo-loaded NPs dimensions if compared with placebo NPs.

**Table 5 pharmaceutics-13-01565-t005:** Characterization of HA-RhB NPs. Results represent the mean ± SD; (*n* = 3 independent batches).

Sample Code	Diameter ± SD (nm)	PDI ± SD	ζ Potential ± SD (mV)
HA-RhB_280	249.76 ± 13.42 ^ns^	0.29 ± 0.06	−16.58 ± 2.21
HA-RhB_540	183.42 ± 25.38 ^ns^	0.30 ± 0.06	−19.70 ± 5.35
HA-RhB_710	218.42 ± 17.06 ^ns^	0.28 ± 0.04	−16.72 ± 2.28
HA-RhB_820	235.15 ± 26.52 ^ns^	0.30 ± 0.02	−19.53 ± 2.16

^ns^ indicates the non-significant difference of HA-RhB NPs dimensions if compared with non-fluorescent NPs.

## Data Availability

The data presented in this study are available on request from the corresponding author.

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
