# Peer review of "Hyaluronic Acid-Based Nanoparticles for Protein Delivery: Systematic Examination of Microfluidic Production Conditions"

_pharmaceutics, 2021, doi:10.3390/pharmaceutics13101565_

Round 1
Reviewer 1 Report
The authors dealt with the synthesis of polymeric NPs for protein delivery using microfluidic device. The selected topic has increasing importance and the results bear significant novelty. Nevertheless, some issues should be considered and corrected prior publication:
In lines 303-304 the authors stated that the HA:CS ratio was kept when the concentration of HA was adjusted to the required viscosity. The consequently changing CS concentration did not affect the viscosity of the solution? Provideing the viscosity curves of the CS solutions for justification would be advantageous.
The discussion of modified flow curves would be better to be placed before the discussion of the NP parameters.
During the discussion of the EE were the average EE% of the different HA samples displayed in line 403?
The authors also stated that PDI was used as selection criteria of best NP samples (line 414), but for Myo 540, the _10 sample had better (0.27) PDI as _15 (0,3), but its EE was only 53%. Nevertheless, the selection criteria should better described/justified. Following a QbD approach and determine a suitable product DS would be advantageous.
Minor comments:
Fig 3. a ) is missing after the text: green dotted line
Fig 4. was separated from its legend
Author Response
Authors would like to thank the reviewers for the constructive suggestions and the opportunity to better implement our paper.
The Manuscript has been revised by the Authors according to the reviewers’ indications.
In the revised version of the Manuscript, changes have been highlighted by using the “Track
Changes” of Word. New versions of Figures and Tables were directly added in the text for better understanding (and uploaded separately).
A point-by-point response to the reviewers' comment suggestions has been generated by reporting Reviewers’ comments (in black) and Authors’ answers (in blue).
Reviewer #1:
The authors dealt with the synthesis of polymeric NPs for protein delivery using microfluidic device. The selected topic has increasing importance and the results bear significant novelty. Nevertheless, some issues should be considered and corrected prior publication:
In lines 303-304 the authors stated that the HA:CS ratio was kept when the concentration of HA was adjusted to the required viscosity. The consequently changing CS concentration did not affect the viscosity of the solution? Providing the viscosity curves of the CS solutions for justification would be advantageous.
Thank you for your positive feedback and valuable recommendations. Table of shear rates/equilibrium flow curve analysis was carried out also for CS solutions. However, varying the CS concentration from 0.125 to 0.043 mg/mL does not affect the solution viscosity. Here below the viscosity curves for the different CS solutions tested. Moreover, the impact of the CS solutions concentration is better pointed out in the revised manuscript. (Figure reported in the file attached)
The discussion of modified flow curves would be better to be placed before the discussion of the NP parameters.
As suggested by the referee, the manuscript has been properly revised.
During the discussion of the EE were the average EE% of the different HA samples displayed in line 403?
The authors thanks for the clarification. In the discussion of Myo loading, the average EE% reached for each HA polymers was used; it was calculated from the EE% of the different Myo:CS for each HA Mw tested. To avoid any misunderstanding, “average EE%” has been added in the revised manuscript and its calculation has been specified.
The authors also stated that PDI was used as selection criteria of best NP samples (line 414), but for Myo 540, the _10 sample had better (0.27) PDI as _15 (0,3), but its EE was only 53%. Nevertheless, the selection criteria should better described/justified. Following a QbD approach and determine a suitable product DS would be advantageous.
The best NPs samples were selected in term of size uniformity (PDI <0.3) and encapsulation efficiency (EE% > 65%). Myo_540_15 batch was selected because it was uniform in size (comparable to Myo_540_20 batch) but it showed the highest Myo EE% (85.90%). The selection criteria and the selected batches were better pointed out in the revised manuscript.
Minor comments:
Fig 3. a ) is missing after the text: green dotted line
The manuscript was revised as suggested by the referee.
Fig 4. was separated from its legend
The manuscript was revised as suggested by the referee.

Reviewer 2 Report
The authors of the paper “Hyaluronic Acid based nanoparticles for protein delivery: systematic examination of microfluidic production conditions” proposed a methodological approach to obtain hyaluronic acid based nanoparticles (HA NPs) as a vehicle for efficient protein delivery using microfluidic technology. The proposed microfluidic based method
The subject addressed by the authors is interesting and in general, the presented results can have a scientific significance. However some improvements are necessary
Here following the critical remarks on specific issues the Authors should address and that could improve their work:
-the use of hMSCs as cell model for biological study should be explain;
-Section 3.3 Fig 6-Why unlike HA_540, the viability (not vitability as in the graph!) of the cells treated with the others HA Mw is affected by NPs concentrations? This should be explaining.
-Section 3.4. It is easier to follow the data by including HA-RhB characteristics into a new table;
-Section 3.4 -Fig. 7 what means “almost” 20 µg of HA-RhB NPs in the caption figure?
-Section 3.4-The uptake experiments were performed after 90 min. How was choose this time point? What about shorter time points? Do the authors investigate the release in the cell medium? This aspect is also important!
-Results and Discussions Section –it seems to me more as data presenting section, with very modest accent on discussions and comments on the results;
- Conclusions-there is a candidate for further studies among the HA NPs tested?
-minor English revision.
Author Response
Authors would like to thank the reviewers for the constructive suggestions and the opportunity to better implement our paper.
The Manuscript has been revised by the Authors according to the reviewers’ indications.
In the revised version of the Manuscript, changes have been highlighted by using the “Track
Changes” of Word. New versions of Figures and Tables were directly added in the text for better understanding (and uploaded separately).
A point-by-point response to the reviewers' suggestions has been generated by reporting Reviewers’ comments (in black) and Authors’ answers (in blue).
Reviewer #2:
The authors of the paper “Hyaluronic Acid based nanoparticles for protein delivery: systematic examination of microfluidic production conditions” proposed a methodological approach to obtain hyaluronic acid based nanoparticles (HA NPs) as a vehicle for efficient protein delivery using microfluidic technology. The proposed microfluidic based method
The subject addressed by the authors is interesting and in general, the presented results can have a scientific significance. However some improvements are necessary
Here following the critical remarks on specific issues the Authors should address and that could improve their work:
-the use of hMSCs as cell model for biological study should be explain;
CD44 is identified as molecular marker for mesenchymal stem cell and it is involved in mesenchymal stem cell requirement in tissue development and regeneration. hMSCs were chosen as model cell line due to the well-established CD44 expression to investigate HA as anchor of attach to CD44 and HA ability to promote the NPs cell internalization by the virtue of the interaction with the HA binding site on the extracellular domain of CD44.
These information and reference (Riva F, Omes C, Bassani R, Nappi RE, Mazzini G, Icaro Cornaglia A, Casasco A. In-vitro culture system for mesenchymal progenitor cells derived from waste human ovarian follicular fluid. Reprod Biomed Online. 2014 Oct;29(4):457-69. doi: 10.1016/j.rbmo.2014.06.006, PMID: 25131558) were added in the revised manuscript.
- Section 3.3 Fig 6-Why unlike HA_540, the viability (not vitability as in the graph!) of the cells treated with the others HA Mw is affected by NPs concentrations? This should be explaining.
Figure 6 has been modified in the revised manuscript.
The authors agree with the referee that the HA/CS NPs concentration affected the cell viability however this viability reduction cannot be defined as toxic since the viable cell percentage is always higher than 80%. So, the increase of NPs concentration slightly affected the overall biological and biochemical cellular functions.
Discussion regarding the NPs concentration effect on the cell viability has been added in the revised manuscript.
- Section 3.4. It is easier to follow the data by including HA-RhB characteristics into a new table.
As suggested by the referee, HA-RhB NPs features have been summarized into a new table (Table 5 in the revised manuscript).
-Section 3.4 -Fig. 7 what means “almost” 20 µg of HA-RhB NPs in the caption figure?
Figure 7 caption has been properly corrected in the revised manuscript.
- Section 3.4-The uptake experiments were performed after 90 min. How was choose this time point? What about shorter time points? Do the authors investigate the release in the cell medium? This aspect is also important!
As reported in literature, HA-coated nanoparticles present an apparent lag phase of about 1 h to be internalized by the cell (Zaki et al, 2011). Based on our previous work (Chiesa et al, 2020), HA/CS NPs uptake in hMSCs has been demonstrated at 90 min of incubation when NPs appear located into the cytosol near the perinuclear region, moreover confocal images taken at 90min of incubation showed that fluorescent HA/CS NPs are confined to discrete cytoplasmic vesicle-like regions, approximately of 1 - 3 μm in diameter, placed near the membrane surface expressing CD44. The authors agree with the referee that prolonging the incubation time a massive NPs uptake should be revealed, however the main purpose of this preliminary test was to verify the effective NPs uptake regardless the different Mw and an incubation of 90 min is enough to discriminate it. Further, more detailed studies are ongoing to investigate the HA/CS NPs cell internalization, the kinetic of the process and the role of HA Mw on the NPs interaction with CD44. Among these the release in the cell medium will be investigated. These studies will be reported in a forthcoming paper as indicated in the Conclusions of the revised manuscript.
References:
- Zaki M. N.; Nasti A.; Tirelli N. Nanocarriers for Cytoplasmic Delivery: Cellular Uptake and Intracellular Fate of Chitosan and Hyaluronic Acid-Coated Chitosan Nanoparticles in a Phagocytic Cell Model. Macromol. Biosci. 2011, 11, 1747–1760. doi: 10.1002/mabi.201100156.
- Chiesa, E.; Riva, F.; Dorati, R.; Greco, A.; Ricci, S.; Pisani, S.; Patrini, M.; Modena, T.; Conti, B.; Genta, I. On-Chip Synthesis of Hyaluronic Acid-Based Nanoparticles for Selective Inhibition of CD44+ Human Mesenchymal Stem Cell Proliferation. Pharmaceutics, 2020, 12, (3):260, doi: 10.3390/pharmaceutics12030260.
-Results and Discussions Section –it seems to me more as data presenting section, with very modest accent on discussions and comments on the results.
Results and Discussions Sections have been implemented in the revised Manuscript.
- Conclusions-there is a candidate for further studies among the HA NPs tested?
In the revised manuscript Conclusions have been better pointed out. So far, there is not a candidate because all the formulations revealed required size and uniformity as well as acceptable Myo encapsulation efficiency. Further studies will deal with the effect of HA Mw on the NPs uptake mechanism with specific focus on the NPs/CD44 interaction and the intracellular pathway involved in their internalization. After these experiments it will be possible to choose the best formulation and these data will be reported in a forthcoming paper.
- minor English revision.
English revision has been performed changes have been highlighting by using the “Track Changes” of Word.

Reviewer 3 Report
Chiesa's et al. manuscript is devoted to the development of polysaccharide-based nanoparticles for the delivery of protein. In general, the manuscript is well structured and illustrated but some important issues should be carefully revised.
- The authors use “Mw” to designate the molecular weight of proteins and polymers. However, Mw means the weight average molecular weight of polymers. Do the authors mean very this? If yes, please designate it in the text. If not, please use “MW” as an abbreviation of molecular weight.
- Introduction. Please, enlarge the information on the known polyelectrolyte interacted nanoparticles. For example, see the papers: Pilipenko et al. Pharmaceutics 2019, 11(7), 317; doi: 10.3390/pharmaceutics11070317; Abdullah, Int. J. Pharm. Investig. 2016, 6(2): 96–105, doi: 10.4103/2230-973X.177823; Yeh et al. Acta Biomater. 2011, 7: 3804-3812, doi: 10.1016/j.actbio.2011.06.026. As for HA-CS nanoparticles, which are already described in the literature (Genarri et al. Beilstein J Nanotechnol. 2019; 10: 2594–2608, doi: 10.3762/bjnano.10.250; etc.), please underline better the novelty of this study.
- What the mechanism of Myo release do the author suppose? It would be nice if the author could add a mathematical treatment (Higuchi, Korsmeyer-Peppas, etc. models) of the release curves and discuss which mechanism they relate to. The value of the paper would increase much more.
- Conclusions. Please, precise conclusions according to the results obtained. What the main tendencies did you revealed in this study? What was the best formulation from the variety of prepared ones?
- Second column of Table 1 and First column of Table 2. Please, remove “kDa” from each line and add this dimension into the column heading as it was done for the dimensions of other parameters given in the neighboring columns (Table 1).
- Improve the quality of Figures 1-4. Please, color the axis and legends in black and increase the font size. Those Figures are barely readable now.
- Increase the size of plots in Figure 5.
Author Response
Authors would like to thank the reviewers for the constructive suggestions and the opportunity to better implement our paper.
The Manuscript has been revised by the Authors according to the reviewers’ indications.
In the revised version of the Manuscript, changes have been highlighted by using the “Track
Changes” of Word. New versions of Figures and Tables were directly added in the text for better understanding (and uploaded separately).
A point-by-point response to the reviewers' suggestions has been generated by reporting Reviewers’ comments (in black) and Authors’ answers (in blue).
Reviewer #3:
Chiesa's et al. manuscript is devoted to the development of polysaccharide-based nanoparticles for the delivery of protein. In general, the manuscript is well structured and illustrated but some important issues should be carefully revised.
- The authors use “Mw” to designate the molecular weight of proteins and polymers. However, Mw means the weight average molecular weight of polymers. Do the authors mean very this? If yes, please designate it in the text. If not, please use “MW” as an abbreviation of molecular weight.
Thanks for the clarification. Weight average molecular weight of HA has been specified and designated as Mw in the revised manuscript. The Myo molecular weight has been designed as MW to avoid any misunderstanding.
- Introduction. Please, enlarge the information on the known polyelectrolyte interacted nanoparticles. For example, see the papers: Pilipenko et al. Pharmaceutics 2019, 11(7), 317; doi: 10.3390/pharmaceutics11070317; Abdullah, Int. J. Pharm. Investig. 2016, 6(2): 96–105, doi: 10.4103/2230-973X.177823; Yeh et al. Acta Biomater. 2011, 7: 3804-3812, doi: 10.1016/j.actbio.2011.06.026. As for HA-CS nanoparticles, which are already described in the literature (Genarri et al. Beilstein J Nanotechnol. 2019; 10: 2594–2608, doi: 10.3762/bjnano.10.250; etc.), please underline better the novelty of this study.
New references have been added in the revised manuscript.
The novelty of this study has been better addressed in revised manuscript. To the best of our knowledge this paper is the first attempt to identify methodological approach to produce HA based NPs through a microfluidic device starting from different HAs but exploiting the same process parameters without impacting the formulation characteristics. This approach could be useful for a rapid product development and an easier scale-up.
- What the mechanism of Myo release do the author suppose? It would be nice if the author could add a mathematical treatment (Higuchi, Korsmeyer-Peppas, etc. models) of the release curves and discuss which mechanism they relate to. The value of the paper would increase much more.
The Myo release data were plotted as (i) a cumulative amount of the drug released versus the time (zero order kinetics model); (ii) ln cumulative percentage of the drug released versus the time (first-order kinetics model); and (iii) a cumulative percentage of the drug release versus the square root of time (Higuchi kinetic model). Moreover, the Myo release data up to 60% were analyzed by the empirical Ritger and Peppas equation.
All the formulations tested satisfied the Higuchi model (R2=0.95) regardless the HA Mw; moreover, the diffusion exponent found by fitting the Myo release data through Ritger and Peppas equation indicates an anomalous diffusion mechanism.
Section regarding in vitro release study has been implemented in the revised manuscript.
- Conclusions. Please, precise conclusions according to the results obtained. What the main tendencies did you revealed in this study? What was the best formulation from the variety of prepared ones?
Conclusions have been better pointed out. So far, there is not the best candidate because all the formulations revealed required size and uniformity as well as acceptable Myo encapsulation efficiency. Further studies will deal with the effect of HA Mw on the NPs uptake mechanism with specific focus on the NPs/CD44 interaction and the intracellular pathway involved in their internalization. After these experiments it will be possible to choose the best formulation and these data will be reported in a forthcoming paper.
- Second column of Table 1 and First column of Table 2. Please, remove “kDa” from each line and add this dimension into the column heading as it was done for the dimensions of other parameters given in the neighboring columns (Table 1).
As suggested by the referee, Table 1 and 2 has been revised.
- Improve the quality of Figures 1-4. Please, color the axis and legends in black and increase the font size. Those Figures are barely readable now.
Figure 1-4 has been improved accordingly with the referee’s suggestions
- Increase the size of plots in Figure 5.
Figure 5 size has been increased.

Round 2
Reviewer 2 Report
The manuscript entitled “Hyaluronic Acid based nanoparticles for protein delivery: systematic examination of microfluidic production conditions”was revised and the authors have made all the recommended modifications.
I agree with the publication of this new, improved form of the manuscript.
Reviewer 3 Report
The authors answered the questions and corrected the manuscript properly.